# Poly(*N*,*N*′-Diethylacrylamide)-Based Thermoresponsive Hydrogels with Double Network Structure

**DOI:** 10.3390/polym12112502

**Published:** 2020-10-27

**Authors:** Lenka Hanyková, Ivan Krakovský, Eliška Šestáková, Julie Šťastná, Jan Labuta

**Affiliations:** 1Department of Macromolecular Physics, Faculty of Mathematics and Physics, Charles University, V Holešovičkách 2, 180 00 Prague 8, Czech Republic; ivank@kmf.troja.mff.cuni.cz (I.K.); eliska6akova@gmail.com (E.Š.); chamky@seznam.cz (J.Š.); 2International Center for Materials Nanoarchitectonics (WPI-MANA), National Institute for Materials Science (NIMS), 1-1 Namiki, Tsukuba, Ibaraki 305-0044, Japan; labuta.jan@nims.go.jp

**Keywords:** thermoresponsive hydrogel, double network, poly(*N*,*N*′-diethylacrylamide), swelling, differential scanning calorimetry, NMR spectroscopy

## Abstract

Temperature response of double network (DN) hydrogels composed of thermoresponsive poly(*N*,*N*′-diethylacrylamide) (PDEAAm) and hydrophilic polyacrylamide (PAAm) or poly(*N*,*N*′-dimethylacrylamide) (PDMAAm) was studied by a combination of swelling measurements, differential scanning calorimetry (DSC) and ^1^H NMR and UV-Vis spectroscopies. Presence of the second hydrophilic network in DN hydrogels influenced their thermal sensitivity significantly. DN hydrogels show less intensive changes in deswelling, smaller enthalpy, and entropy changes connected with phase transition and broader temperature interval of the transition than the single network (SN) hydrogels. Above the transition, the DN hydrogels contain significantly more permanently bound water in comparison with SN hydrogels due to interaction of water with the hydrophilic component. Unlike swelling and DSC experiments, a rather abrupt transition was revealed from temperature-dependent NMR spectra. Release study showed that model methylene blue molecules are released from SN and DN hydrogels within different time scale. New thermodynamical model of deswelling behaviour based on the approach of the van’t Hoff analysis was developed. The model allows to determine thermodynamic parameters connected with temperature-induced volume transition, such as the standard change of enthalpy and entropy and critical temperatures and characterize the structurally different states of water.

## 1. Introduction

Stimuli-responsive polymer hydrogels are great interest for advanced applications as smart materials, because of the ability to change their properties (volume, transparency, adsorption) in response to different external stimuli, such as temperature, pH, presence of chemicals, exposure to light, electromagnetic field [1,2,3,4,5]. Among various stimuli, the temperature is the most extensively exploited in the field of ‘smart’ polymers due to the important role of temperature in nature [6,7,8,9,10,11].

Thermosensivity of polymer hydrogels is associated with a changed balance between various types of interactions, mainly hydrogen bonds and hydrophobic interactions. At temperatures below volume phase transition temperature (VPTT) hydrogels absorb water to reach the swollen state and above VPTT they release water and shrink. On the molecular level, volume phase transition (collapse) in crosslinked hydrogels is assumed to be a macroscopic manifestation of a coil-globule transition, as was shown for poly(*N*-isopropylacrylamide) (PNIPAm, VPTT ≈307 K) in water by light scattering [12].

Interpenetrating and semi-interpenetrating networks represent promising composite materials as they allow combining different polymers either to enhance characteristics of its components or to obtain materials with unique properties. Recently, novel concept of double network (DN) hydrogels as a class of interpenetrating polymer networks composed of two highly asymmetrically crosslinked networks swollen in water was introduced by Gong [13,14]. The excellent mechanical performances of DN hydrogels originate from this asymmetric combination of two networks when upon deformation, internal fractures in the first network are formed and these fractures act as additional crosslinkers reinforcing the hydrogel [15,16]. Based on the DN principle, several novel systems with enhanced mechanical property and techniques were developed such as DN hydrogels with controlled shape deformation [17], organic/inorganic DN hydrogel composites [18], or flexible display devices based on DN hydrogels [19].

Several other DN hydrogels have been reported but virtually only a few of them where thermoresponsive. DN PNIPAm/PNIPAm hydrogels containing inorganic polysiloxane nanoparticles [20] or comprised of ionized first network with electrostatic comonomer [21] were studied. The influence of the hydrogel composition on volume phase transition, morphology, equilibrium swelling, deswelling-reswelling kinetics, and mechanical properties was evaluated. PNIPAm DN hydrogels were also formed as micropillar arrays [22] or membranes [23]. pH- and temperature-responsive DN hydrogel based on PNIPAm and polyacrylic acid using graphene oxide as an additive were prepared and the effect of additive and acid contents on various physical properties was investigated [24,25].

The critical temperature in the physiological range together with the fact the PNIPAm is a well-understood and well-characterized polymer has contributed to the popularity of PNIPAm hydrogels. However, in connection with the application of PNIPAm, some concerns appeared around potential release of toxic low-molecular-weight amines due to hydrolysis. Therefore, other thermoresponsive polymers have been proposed as alternatives for bioapplications [26]. Among them, poly(*N*,*N*′-diethylacrylamide) (PDEAAm) with VPTT values ≈304–307 K is considered suitable for applications in the life sciences as cytotoxicity of PDEAAm hydrogels is less pronounced than for PNIPAm ones [27].

In the present work, we use the concept of DN to design and characterized thermoresponsive hydrogels composed of two contrasting polymer networks, a tightly crosslinked first network of temperature-sensitive PDEAAm and a loosely crosslinked second hydrophilic network of polyacrylamide (PAAm) (DN hydrogels DNA) or poly(*N*,*N*′-dimethylacrylamide) (PDMAAm) (DN hydrogels DNM). A series of DN hydrogels with various crosslinking densities of the first network and a constant crosslinking density of the second network was prepared. Swelling measurements, differential scanning calorimetry (DSC), NMR and UV-Vis spectroscopies were combined to provide information about temperature induced volume changes in hydrogels on macroscopic and molecular scales. The behaviour of the prepared DN hydrogels is compared with the behaviour of corresponding reference single network (SN) hydrogels containing PDEAAm.

## 2. Materials and Methods

### 2.1. Hydrogels Synthesis

SN hydrogels were prepared by redox polymerization of aqueous solutions containing monomer, *N*,*N*′-diethylacrylamide (DEAAm, Sigma-Aldrich, St. Louis, MO, USA), crosslinking agent, *N*,*N*′-methylenebisacrylamide (MBAAm, Sigma-Aldrich, St. Louis, MO, USA), initiator, ammonium persulfate (APS, Sigma-Aldrich, St. Louis, MO, USA), and catalyst, *N*,*N*,*N*′,*N*′-tetramethylenediamine (TEMED, Sigma-Aldrich, St. Louis, MO, USA). All ingredients, except for TEMED, were dissolved in deionized water and flushed with nitrogen for ca 10 min, after which TEMED was added. Approximately 3 mL of solution was injected by a syringe in a mold assembled from two glassy plates separated by a 1 mm spacer from silicone rubber with a rectangular void of dimensions 7 × 5 cm^2^. The polymerization proceeded in a refrigerator at 5 °C for 24 h. After preparation, hydrogel samples were washed by a large amount of distilled deionized water for 1 day. Washing was repeated three times to remove residual unreacted reagents. Compositions of reaction mixtures used in the preparation of SN hydrogels are given in Table 1.

DNA double network hydrogels were prepared as follows: Specimens cut from SN hydrogels at 20 °C having dimensions ca 3 × 3 × 0.1 cm^3^ were immersed for 24 h in a large volume of aqueous solution containing monomer, 142.2 g.L^−1^ acrylamide (AAm), crosslinking agent, 0.15 g.L^−1^ MBAAm, and photoinitiator, 0.15 g.L^−1^ 2-oxoglutaric acid (OGA) at the same temperature. Before the swelling procedure, the swelling solution was flushed with nitrogen for ca 30 min. DN hydrogels were prepared at room temperature (ca 20 °C) by UV irradiation (365 nm, 3 h) of the swollen specimen fixed between two glassy plates separated by a spacer from silicone rubber of a proper thickness with a rectangular void of dimensions 4 × 4 cm^2^. After preparation, hydrogel samples were washed three times by a large amount of distilled water to remove residual unreacted reagents. DNM double network hydrogels were prepared in the same way with aqueous solution containing monomer, 198.26 g.L^−1^
*N*,*N*′-dimethylacrylamide (DMAAm), crosslinking agent, 0.15 g.L^−1^ MBAAm, and photoinitiator, 0.15 g.L^−1^ 2-oxoglutaric acid (OGA). The concentrations of monomers (AAm and DMAAm) were chosen to prepare both DN hydrogels in the same molar ratio of monomers, i.e., DEAAm:AAm = 1:2 and DEAAm:DMAAm = 1:2.

The numbering of DNA and DNM samples refers to that of SN samples used in their preparation, e.g., DNA1 hydrogel was prepared from single network hydrogel SN1. Each network sample (SN1, SN2, SN3, DNA1, DNA2, etc.) was independently synthesized in two specimens. All experiments were performed on both specimens to probe repeatability and reliability of obtained characteristic parameters. Examples of repeated swelling and NMR experiments for hydrogel DNA1 are shown in Appendix A.

### 2.2. Differential Scanning Calorimentry

The calorimetric measurements were performed using a DSC8500 apparatus (Perkin-Elmer, Waltham, MA, USA). Purge gas (nitrogen) was let through the DSC cell with a flow rate of 20 mL/min. The temperature of the equipment was calibrated with mercury, distilled water, and indium. The melting heat of indium was used for calibrating the heat flow.

Specimen of mass ca 30 mg were cut from hydrogels at room temperature (ca 20 °C) and transferred quickly to DSC pans. They were subjected to a first heating scan from 20 °C to 60 °C, held at 60 °C for 1 min, subjected to a first cooling scan to 20 °C, held at 20 °C for 5 min, subjected to a second heating scan from 20 °C to 60 °C, held at 60 °C for 1 min and, finally, subjected to a second cooling scan from 60 °C to 20 °C. All scans were carried out at a rate of 10 °C/min. Higher heating/cooling rates were chosen to obtain a reasonable signal from DN hydrogels.

The values of onset temperature of demixing $T_{\mathrm{DSC}}^{\mathrm{on}}$ were determined from the intersection of the baseline and the leading edge of the exotherm. The values of specific enthalpy of demixing ∆*H*_DSC-ms_ (in units of Joule per one gram of mass of the sample) and ∆*H*_DSC-pn_ (in units of Joule per gram of polymer network) were calculated by integration of demixing endotherms obtained in the second heating.

### 2.3. ^1^H NMR Spectroscopy

^1^H NMR measurements were performed with a Bruker Avance 500 liquid-state spectrometer (Bruker, Karlsruhe, Germany) operating at 500.1 MHz. Typical conditions were as follows: π/2 pulse width of 12.5 μs, relaxation delay of 20 s, spectral width of 5 kHz, acquisition time of 1.64 s, and 16 scans. The integrated intensities were determined by spectrometer integration software with an accuracy of ±1%. Constant temperature within ±0.2 K was maintained in all measurements using a BVT 3000 temperature unit. The samples were always kept at the experimental temperature for 15 min before the measurement.

To quantitatively characterize the phase transition, we have used the values of the collapsed *p*-fraction (degree of collapsing) obtained as
(1)$$p(T)=1-\frac{I}{I_{0}}$$
where *I* is the integrated intensity of the given polymer signal in the spectrum of partly collapsed hydrogel and *I*_0_ is the integrated intensity of this signal if no collapse transition occurs. For *I*_0_, we took values based on integrated intensities as obtained at 25 °C and taking into account the fact that the integrated intensities should decrease with absolute temperature as 1/*T* [28,29]. Temperature dependences of *p*-fraction were fitted using equation [29,30]
(2)$$p\left(T\right)=\frac{p_{\max}}{1\;+\;\exp\left(\frac{\Delta H_{\mathrm{NMR}}}{RT}\;-\;\frac{\Delta S_{\mathrm{NMR}}}{R}\right)}$$
where *p*_max_ is the maximum value of *p*-fraction, Δ*H*_NMR_ and Δ*S*_NMR_ are the enthalpy and entropy changes connected with the phase transition, *R* is the gas constant and *T* is the absolute temperature. Values of NMR-determined transition temperatures, $T_{\mathrm{NMR}}^{\mathrm{on}}$ were obtained as the onset temperature of the sigmoidal *p*(*T*) curve constructed as an intersecting point between tangent at the inflexion point of *p*(*T*) and *x*-axis. Similarly, the width of the transition Δ*T*_NMR_ was determined as the difference between offset and onset temperature [30].

The ^1^H spin-spin relaxation times *T*_2_ of hydrogen-deuterium oxide (HDO) were measured using the same instrument and the CPMG [31] pulse sequence 90°_x_-(*t*_d_-180°_y_-*t*_d_)_n_-acquisition with *t*_d_ = 0.5 ms, relaxation delay 100 s, and 8 scans. The total time for *T*_2_ relaxation was an array of 35 values.

### 2.4. UV-Vis Experiments

Release behaviour of SN and DN hydrogels was studied using methylene blue (MB) solution (Sigma-Aldrich, St. Louis, MO, USA). The pieces of hydrogels (≈2 cm^2^) were dried at room temperature for three days and then in a vacuum oven until it reached the constant weight. Dried hydrogels were immersed in MB solution (0.15 g/L) at room temperature and left for two days to attain equilibrium swelling condition. The concentration of MB molecules inside hydrogels *c*_0_ was determined from the mass of dried and swollen hydrogels. The hydrogels were then removed from MB solution to phials filled with 4 mL of water placed in water bath thermostated at 45 °C. Time-dependence of MB concentration was detected with UV-Vis spectrophotometer (Hitachi U-2900, Hitachi, Tokyo, Japan) at 664 nm using MB signal of maximum intensity at 664 nm. At regular time intervals, MB solution released from hydrogel was pipetted out and the concentration of MB *c*(*t*) was measured. The fraction of released MB *f*(*t*) was calculated as
(3)$$f\left(t\right)=\frac{c(t)}{c_{0}}$$

Obtained time-dependencies of the fraction of released MB *f*(*t*) was fitted with expression
(4)$$f\left(t\right)=f_{\mathrm{MB}}-b\exp\left(-\frac{t}{\tau }\right)$$
where *f*_MB_ is the equilibrium fractions of released MB, *τ* is the release delay time and *b* is a coefficient.

### 2.5. Swelling Behaviour

A cylindrical specimen of diameter ca 10 mm and height ca 1–3 mm was cut from hydrogel samples prepared and swollen to equilibrium at room temperature (ca 25 °C) in bottles containing ca 50 mL distilled deionized water. Then, the bottles were transferred to a thermostated bath kept at a starting temperature (20 °C) and equilibrated for 2 hrs. Masses of swollen samples were measured using a precise balance. Then, the bottles containing hydrogel samples and water were heated to a next temperature and equilibrated again for 2 hrs. Measurements at progressively increasing temperatures until temperature 60 °C was achieved were carried out in the same way.

To determine the swelling ratio of hydrogels at 25 °C, hydrogel samples swollen at 25 °C were dried first in the open air at room temperature for 1 day followed by 1 day drying in vacuo at a temperature 80 °C. The swelling ratio of swollen samples at 25 °C, *r*(*T*_0_) (hereinafter denoted as α), was calculated by
(5)$$\alpha =r\left(T_{0}\right)=\frac{m^{\prime }\left(T_{0}\right)}{m_{\mathrm{dry}}}$$
where *m’*(*T*_0_) and *m*_dry_ are the masses of sample swollen at 25 °C and dry sample, respectively. Then, the swelling ratio in samples swollen at a temperature *T* > *T*_0_, *r*(*T*), was calculated by
(6)$$r\left(T\right)=\frac{m^{\prime }\left(T\right)}{m_{\mathrm{dry}}}$$
where *m’* (*T*) is the mass of hydrogel sample swollen at temperature *T*.

### 2.6. Thermodynamic Model of Deswelling

Thermodynamic model of gravimetric deswelling experiments of SN and DN is schematically shown in Figure 1a. The approach of building this model is similar to the models describing phase separation in solutions of linear polymers studied by NMR spectroscopy [29,30]. Below transition temperature (*T* ≤ *T*_0_), the water molecules are bound to the network, and upon temperature increase, the water is expelled while the network shrinks. The process ends with a collapsed network containing some amount of permanently bound water. Note that (absolute) temperature *T*_0_ is chosen slightly below any collapse of the network can be observed. The whole process is modelled as a transition between two states (swollen and collapsed) (Figure 1b) and described by equilibrium constant *K,* as shown in Equation (7).
(7)$$K=\frac{n_{\mathrm{fw}}}{n_{\mathrm{bw}}}=\frac{m_{\mathrm{fw}}}{m_{\mathrm{bw}}}$$
where *n*_fw_ and *n*_bw_ are the number of moles of free water and bound (but releasable) water molecules, respectively. Using formula *m* = *n*·*M*_w_, where *m* is the mass of water molecules, *n* is the number of moles of water molecules, and *M*_w_ is the molar mass of water, the right-hand side term of the Equation (7) can be obtained. Thus, the equilibrium constant connects experimentally observable quantities of the mass of free (*m*_fw_) and bound (*m*_bw_) but releasable water.

The total amount of water *m*_totw_ contained in the entire system can be expressed as Equation (8).
(8)$$m_{\mathrm{totw}}=m_{\mathrm{fw}}\left(T\right)\;+\;m_{\mathrm{bw}}\left(T\right)\;+\;m_{\mathrm{pbw}}$$
where *m*_fw_(*T*), *m*_bw_(*T*) and *m*_pbw_ are masses of free water, bound (but releasable) water and permanently bound water molecules, respectively. Note that only the *m*_fw_(*T*) and *m*_bw_(*T*) terms are temperature dependent. The mass of the network sample at temperature *T* is given by Equation (9).
(9)$$m^{\prime }\left(T\right)=m_{\mathrm{dry}}\;+\;m_{\mathrm{pbw}}\;+\;m_{\mathrm{bw}}\left(T\right)$$
where *m*_dry_ is the mass of dry polymer network. Note that the quantity *m’*(*T*) in the model represents the mass of the network and contained water (at *T*_0_ it is the entire system and at *T* > *T*_0_ it is a part of the system bounded within the dashed-line square) (Figure 1a). The quantities denoted by prime are related to the polymer network (with all contained water) and not to the entire system. Using Equations (8) and (9) the mass of the network sample at *T* = *T*_0_ is given by Equation (10).
(10)$$m^{\prime }\left(T_{0}\right)=m_{\mathrm{dry}}\;+\;m_{\mathrm{totw}}$$Expressing *m*_bw_(*T*) from Equation (9) and *m*_totw_ from Equation (10) with subsequent substitution into Equation (8) yields a formula for the mass of free water in the form of Equation (11).
(11)$$m_{\mathrm{fw}}\left(T\right)=m^{\prime }\left(T_{0}\right)-m^{\prime }\left(T\right).$$ Formulas for free (*m*_fw_(*T*)) and bound (*m*_bw_(*T*)) masses of water as obtained from Equation (11) and rearrangement of Equation (9), respectively, lead to the expression of the equilibrium constant *K* (Equation (7)) as a function of actual mass of sample *m’*(*T*) at temperature *T* with other parameters (*m’*(*T*_0_), *m*_dry_ and *m*_pbw_) as constants connected with each polymer network type and design (Equation (12)).
(12)$$K=\frac{m^{\prime }\left(T_{0}\right)-m^{\prime }\left(T\right)}{m^{\prime }\left(T\right)-m_{\mathrm{dry}}-m_{\mathrm{pbw}}}$$ After rearrangement, the *m’*(*T*) can be expressed from Equation (12) in the form of Equation (13).
(13)$$m^{\prime }\left(T\right)=\frac{m^{\prime }\left(T_{0}\right)\;+\;\left(m_{\mathrm{dry}}\;+\;m_{\mathrm{pbw}}\right)\;K}{1\;+\;K}$$ The temperature behaviour of hydrogels is monitored as a swelling ration *r*(*T*) defined by Equation (6) as a ratio of the swollen mass of the network sample to the mass of dry network. Using Equation (13) for *m’*(*T*), the following formula for *r*(*T*) is obtained.
(14)$$r\left(T\right)\;=\frac{m^{\prime }\left(T_{0}\right)/m_{\mathrm{dry}}\;+\;\left(1\;+\;m_{\mathrm{pbw}}/m_{\mathrm{dry}}\right)\;K}{1\;+\;K}$$ The equilibrium constant *K* is connected to Gibbs free energy change Δ*G* by Equation (15).
(15)$$K=e^{-\;\frac{\Delta G_{\mathrm{grav}}}{RT}}=e^{-\;\frac{\Delta H_{\mathrm{grav}}-T\Delta S_{\mathrm{grav}}}{RT}}$$
where Δ*G*_grav_ = Δ*H*_grav_ ‒ *T*Δ*S*_grav_, and Δ*H*_grav_ and Δ*S*_grav_ are the standard change of enthalpy and entropy, respectively, connected with the collapse of the network as obtained from gravimetric measurement. The units of Δ*H*_grav_ and Δ*S*_grav_ are Joule per mol of cooperative water units and Joule per Kelvin per mol of cooperative water units, respectively (the details are given in the text below). After the substitution of *K* from Equation (15) into Equation (14) and denoting the rations *m’*(*T*_0_)/*m*_dry_ as *α* (see Equation (5)) and *m*_pbw_/*m*_dry_ as *β* the final equation for swelling ration *r*(*T*) is obtained in the form of Equation (16).
(16)$$r\left(T\right)=\frac{\alpha \;+\;\left(1\;+\;\beta \right)\;e^{-\;\frac{\Delta H_{\mathrm{grav}}-T\Delta S_{\mathrm{grav}}}{RT}}}{1\;+\;e^{-\;\frac{\Delta H_{\mathrm{grav}}-T\Delta S_{\mathrm{grav}}}{RT}}}$$
where
$$\alpha =m^{\prime }\left(T_{0}\right)/m_{\mathrm{dry}}$$
$$\beta =m_{\mathrm{pbw}}/m_{\mathrm{dry}}$$

The features of the *r*(*T*) curve, including the definition of onset temperature of swelling $T_{\mathrm{grav}}^{\mathrm{on}}$ and width of the transition region $\Delta T_{\mathrm{grav}}$ are shown in Figure 2. The parameters *α* and *β* are characteristic for each polymer network with physical meaning of maximum swelling ratio (at *T*_0_) and the ratio of permanently bound water to mass of dry network, respectively. It is also of interest to evaluate the ratio of the mass of permanently bound water to the total mass of water contained in a fully swollen network (at *T* = *T*_0_) denoted as *γ* = *m*_pbw_/*m*_0_. Using *α*, *β* and Equation (10) the *γ* = *β*/(*α* ‒ 1). The *r*(*T*), as expressed in Equation (16), is fitted to experimental data. The parameter *α* is experimentally determined from the first point of *r*(*T*) curve (at *T* = *T*_0_), the parameters *β*, Δ*H* and Δ*S* are fitted and *γ* is then calculated using *α* and *β*.

The enthalpy change Δ*H*_grav_ as obtained from Equation (16) using the fitting procedure is called “van’t Hoff transition enthalpy” and has a useful connection with the so-called cooperative unit and its size [32,33,34]. In general, the cooperative unit consists of cooperating molecules (polymer chains or their parts and water molecules) that change their state simultaneously [33]. During the phase transition, the polymer chains collapse, and water molecules are released from the polymer network (except those permanently bound) as seen in Figure 3. During gravimetric measurement, water released from the sample is observed, therefore the intrinsic unit of van’t Hoff transition enthalpy Δ*H*_grav_ is Joule per mol of cooperative water units. A cooperative water unit is a water contained in one cooperative unit, which is released during the collapse of the unit. The number of water molecules in a cooperative water unit *N*_wcu_ is given by as a ratio of van’t Hoff transition enthalpy Δ*H*_grav_ and calorimetric enthalpy Δ*H*_DSC_/*C*_rw_ as shown in Equation (17) [32,33,34].
(17)$$N_{\mathrm{wcu}}=\frac{\Delta H_{\mathrm{grav}}}{\Delta H_{\mathrm{DSC}-\mathrm{ms}}/C_{\mathrm{rw}}}$$ The *C*_rw_ normalization constant is introduced in order to have denominator Δ*H*_DSC_/*C*_rw_ with the dimension of Joule per mol of released water molecules. The Δ*H*_DSC-ms_ has a unit of Joule per gram of network sample. The *C*_rw_ is defined as a number of mols of water molecules released during the collapse of one gram of the network sample and can be expressed as Equation (18).
(18)$$C_{\mathrm{rw}}=\frac{n_{\mathrm{fw}}\left(T\to \infty \right)}{m^{\prime }\left(T_{0}\right)}$$
where $n_{\mathrm{fw}}\left(T\to \infty \right)$ is the number of moles of free water at a temperature far above the phase transition temperature and can be obtained from gravimetric measurement as $n_{\mathrm{fw}}\left(T\to \infty \right)=\;m_{\mathrm{fw}}\left(T\to \infty \right)/M_{w}$. Where the mass of free water at a temperature far above the phase transition temperature $m_{\mathrm{fw}}\left(T\to \infty \right)$ is equal to the mass of bound water at *T* = *T*_0_, i.e., $m_{\mathrm{fw}}\left(T\to \infty \right)$ = $m_{\mathrm{bw}}\left(T_{0}\right)$ (see Figure 1a and Equation (8)). From Equation (9) the $m_{\mathrm{bw}}\left(T_{0}\right)$ term can be obtained as $m_{\mathrm{bw}}\left(T_{0}\right)=\;m^{\prime }\left(T_{0}\right)-m_{\mathrm{dry}}-m_{\mathrm{pbw}}$. Then the *C*_rw_ normalization constant has a form of Equation (19).
(19)$$C_{\mathrm{rw}}=\frac{m^{\prime }\left(T_{0}\right)-m_{\mathrm{dry}}-m_{\mathrm{pbw}}}{M_{w}\;m^{\prime }\left(T_{0}\right)}$$ Using the expressions for *α* and *β* (see Equation (16)) the *C*_rw_ has the form of Equation (20).
(20)$$C_{\mathrm{rw}}=\frac{\alpha \;-\;\left(1\;+\;\beta \right)}{\;\alpha M_{W}}$$ The resulting number of water molecules in a cooperative water unit *N*_wcu_ is given by Equation (21).
(21)$$N_{\mathrm{wcu}}=\frac{\Delta H_{\mathrm{grav}}}{\Delta H_{\mathrm{DSC}-\mathrm{ms}}}\frac{\alpha \;-\;\left(1\;+\;\beta \right)}{\;\alpha M_{W}}$$

## 3. Results and Discussion

### 3.1. Swelling Behaviour

The swelling curves for hydrogels SN2, DNA2, and DNM2 (the swelling ratio of hydrogels vs. temperature) during heating are shown in Figure 4. (for other hydrogels see Appendix A) A continuous deswelling of the hydrogels with increasing temperature is observed. For both DN hydrogels DNA2 and DNM2, presence of the second hydrophilic network (PAAm and PDMAAm, respectively) has a strong impact on their temperature dependence of swelling behaviour. The transition region from the expanded (at low temperatures) to collapsed (at high temperatures) state is slightly shifted to higher temperatures in DN networks, the steepness of the transition region decreases and magnitude of deswelling of the DN hydrogels is strongly reduced relative to corresponding SN hydrogel.

The swelling dependences were fitted using Equation (16) and the transition parameters are listed in Table 2. It is evident from Table 2 that hydrogels with the smallest network density (SN3, DNA3, and DNM3) have significantly higher initial swelling ratio (parameter *α*) than hydrogels which are crosslinked more densely. As it further follows from Table 2, DNM hydrogels show up to 2.5 times higher initial swelling ratio than DNA hydrogels. This is unexpected behaviour if we consider that PDMAAm units contain hydrophobic methyl groups in their structure.

Much stronger deswelling of SN hydrogels with temperature is demonstrated by their higher variations of enthalpy Δ*H*_grav_ and entropy Δ*S*_grav_ in comparison with DN hydrogels. At the same time, the presence of the second, temperature-insensitive and hydrophilic component increases the temperature onset $T_{\mathrm{grav}}^{\mathrm{on}}$ and interval of the transition $T_{\mathrm{grav}}^{\mathrm{on}}$ of DN hydrogels.

The most pronounced transition was detected for SN3 and SN2 with the lower crosslinking density. For more densely crosslinked hydrogels SN1 the deswelling is of smaller extent. It means that in SN hydrogels that are crosslinked less, conditions for volume phase transition become to be satisfied. This is reflected in higher variations of enthalpy and entropy of SN3 and SN2 and a narrower transition temperature range.

On the other hand, the entropies and enthalpies for double networks DNA (PDEAAm/PAAm) and DNM (PDEAAm/PDMAAm) decrease with the higher crosslinking density of PDEAAm network. The most likely explanation is related to the process of the preparation, during which the final composition of DN hydrogels changes for different network densities. The SN hydrogels of PDEAAm with higher network density probably swell a lesser number of monomers AAm or DMEAAm during the polymerization of the second component than those with lower network density. That leads to the dependence between the extent of the collapse described with entropy and enthalpy and the network density, as the more hydrophilic component is contained, the less intensive transition occurs.

It is evident from Table 2 that in comparison with SN hydrogels, DN hydrogels show smaller enthalpy and entropy change connected with the phase transition, temperature interval of the transition is broader, and the onset temperature is shifted to higher values. The second hydrophilic component prevents PDEAAm units from changing conformation, and thus DN hydrogels are less sensitive to temperature. Above the transition, the DN hydrogels contain significantly more permanently bound water in comparison with SN hydrogels (ratio *γ = m*_pbw_/*m*_0_ in Table 2) due to interaction of water with hydrophilic component even at elevated temperatures. These observations agree well with results obtained on DN hydrogels composed of PNIPAm and PAAm [35].

### 3.2. Differential Scanning Calorimetry

Examples of DSC thermograms as obtained for SN1 and DNA1 hydrogels are shown in Figure 5. The values of onset and peak temperatures of demixing, specific enthalpy of demixing and specific enthalpy per unit mass of polymer obtained in the second heating cycle are summarised in Table 3.

One rather broad endothermic transition was observed in all SN hydrogels investigated in this study (for SN1 see Figure 5; all data for SN hydrogels are shown in Appendix A). Previously, broad DSC peak was detected for PDEAAm solutions [36] and PDEAAm IPN hydrogels [37], and we suppose that a significant peak broadness (≈15 °C) is related to the chemical structure of the DEAAm side chains. For DN hydrogels, the DSC thermograms (Figure 5; for all data, see Appendix A) are even broader, and the onset temperatures of demixing are about 1 °C higher than in corresponding SN hydrogels. The onset temperatures as detected by DSC for all studied hydrogels are much higher than deswelling temperatures (ca 4–12 °C). Endotherms registered for DN hydrogels are much less distinct than in SN hydrogels.

Specific enthalpy of demixing Δ*H*_DSC-ms_ (per mass of sample) decreases with decreasing crosslinking density in both, SN and DN hydrogels. However, when related to polymer content, specific enthalpy of demixing per unit mass of polymer network, Δ*H*_DSC-pn_, is almost constant in SN hydrogels (≈20 J.g^−1^). These observations agree well with trends and values obtained on DN PNIPAm hydrogels [35]. In DN hydrogels, Δ*H*_DSC-pn_ values are significantly reduced as hydrophilic component in addition to temperature-sensitive PDEAAm is present in the polymer structure. Δ*H*_DSC-pn_ values of PDEAAm/PAAm or PDEAAm/PDMAm hydrogels represent ca 10% or 1%, respectively, of Δ*H*_DSC-pn_ determined for SN hydrogels of PDEAAm.

Based on the approach discussed in Section 2.6 and Equations (18)–(21), the number of water molecules in a cooperative water unit *N*_wcu_ were calculated and summarized in Table 3. The hydrogels SN1 and SN2 with the highest calorimetrical enthalpy changes show the smallest values of *N*_wcu_. One can suppose that the interactions between water molecules within the cooperative unit are enhanced (Figure 3) and thus, the interaction of water–polymer could be weakened. Consequently, a bigger cooperative water unit is more easily expelled from hydrogel than a smaller cooperative unit having stronger interactions with the polymer.

### 3.3. NMR

Temperature-variable high-resolution ^1^H NMR spectra for all hydrogels are included in Appendix A, Appendix A. As an example, Figure 6 shows NMR spectra of hydrogels SN1 and DNA1 measured under the same conditions at 25 °C and 39 °C, i.e., below and above the transition temperature. The assignment of resonances to corresponding proton types is shown in the spectra measured at 25 °C. The strong peak A corresponds to water signal (HDO). Peaks B and E correspond to ethyl CH_2_ and CH_3_ groups of PDEAAm, respectively. Main chain CH groups from components PDEAAm and PAAm have signals C and C´, respectively, signal D corresponds to main chain CH_2_ group of both components. The most important effect observed in the NMR spectra measured at higher temperature is a marked decrease in the integrated intensity of all PDEAAm signals. The mobility of most PDEAAm units is significantly reduced to such an extent that the corresponding signals are too broad to be detected in high-resolution NMR spectra [35,37,38]. On the other hand, signals C´ and D in the spectra of DN hydrogels (Figure 6b) practically do not change with temperature. Taking into consideration that major contribution to these signals comes from CH_2_ and CH units of PAAm we can conclude that the hydrophilic PAAm units do not participate in collapse transition.

Figure 7 shows the temperature dependences of the *p*-fraction as obtained for methyl CH_3_ signals in PDEAAm units (signal E) in SN and DNA hydrogels (for data on DNM see Appendix A). *p*(*T*) dependences were fitted using Equation (2) and the fitted transition parameters are listed in Table 4. It is evident from Table 4 and Figure 7 that transition temperature $T_{\mathrm{NMR}}^{\mathrm{on}}$ practically does not depend on crosslinking density for both SN and DN hydrogels. With this respect, the composition of hydrogels influences $T_{\mathrm{NMR}}^{\mathrm{on}}$ more significantly than crosslinking density. As it was found for IPNs PDEAAm/PAAm [37], critical temperature $T_{\mathrm{NMR}}^{\mathrm{on}}$ linearly decreased with the content of PDEAAm units. In comparison with transition parameters obtained from swelling experiments (Table 2), NMR-determined width of transition Δ*T*_NMR_ is lower and enthalpy Δ*H*_NMR_ and entropy Δ*S*_NMR_ have considerably larger values. This indicates a strongly cooperative dehydration process during the phase transition. The two methods (swelling and NMR) detect different aspects of the transition. Swelling experiments register release of water molecules from hydrogel while the aggregation of hydrogel polymer units due to temperature-induced interaction changes followed by water releasing process is detected by NMR spectroscopy. Expelling of water molecules is a time-consuming process and leads to broad transition interval $\Delta T_{\mathrm{grav}}$ and small values of enthalpy Δ*H*_grav_ particularly for DN hydrogels where hydrophilic component even prevents and slows down this process. On the other hand, as appropriate thermodynamic and interaction conditions are established, PDEAAm units collapsed into impact globular-like structures, and this conformational change process is relatively fast.

Table 4 contains ratios of the two components in samples of DNs as determined at room temperature from integrated intensities of ^1^H NMR signals C‘ and D corresponding to the main chain CH-CH2 protons of PDEAAm and PAAm (PDMAAm) units and signal E corresponding to CH_3_ protons of PDEAAm (cf. Figure 6b). These values are practically 2–4 times higher than the composition of reaction mixtures which was established during the synthesis of DN hydrogels (PDEAAm:PAAm monomer molar ratio = 1:2 and PDEAAm:PDMAAm monomer molar ratio = 1:2, see Section 2.1). This discrepancy could be explained as a possible formation of heterogeneous structure in DN hydrogels as we reported for PNIPAm/PAAm hydrogels [35]. It follows that as a consequence of the preparation process, a great part of PDEAAm could be involved in agglomerates and mobility of PDEAAm units is thus significantly reduced. These immobile units do not contribute to high-resolution ^1^H NMR spectra (the signal is too broad to be detectable) even at temperatures below transition which increases NMR detected molar ratio of AAm or DMAAm compared to DEAAm in DN hydrogels *N*_A_*/N*_DEAAm_ or *N*_M_*/N*_DEAAm_, respectively. Therefore, these two values do not match the ratio of monomers as used in synthesis. A part of PDEAAm units which remain mobile at lower temperatures, during subsequent heating exhibit further agglomeration and the integrated intensity of corresponding NMR signals thus decreases. We suppose that change of hydrated state of mainly these PDEAAm units is detected by NMR, and therefore rather intensive transition with relatively high maximum values of *p*-fraction ≈0.9–1 is revealed by NMR, while considerably small enthalpy values were detected by DSC for all DN hydrogels (cf. Table 3).

### 3.4. NMR Relaxation

Dynamical behaviour of water molecules during the transition was studied using measurements of NMR spin-spin relaxation time *T_2_* on HDO signal. Single-exponential relaxation decay characterized by single relaxation *T_2_* was detected for all hydrogels at temperature 17 °C below the transition. Heating above the transition (45 °C) leads to bi-exponential relaxation decay for all SN hydrogels and double network hydrogel DNA1. The *T_2_* values as obtained for all hydrogels are summarised in Table 5. As it follows from Table 5, *T*_2_ components of bi-exponential dependences differ by orders of magnitude and they can be marked as “confined” and “free” water. “Confined” water molecules are obviously located in collapsed deswollen domains while “free” water molecules are included in less deswollen domains or they are expelled from hydrogel interior. In relation to the schematic model in Figure 1, “confined” water could be assigned to permanently bound water molecules with restricted mobility.

*T*_2_ values for “confined” water in collapsed SN and DNA1 hydrogels are up to two orders of magnitude smaller in comparison with “free” water molecules. The main source of these differences is evidently the fact that the motion of “confined” water is spatially restricted and anisotropic [39]. The occurrence of “confined” water molecules with slow motion was previously detected also in collapsed poly(vinyl methyl ether) and PNIPAm hydrogels [40,41,42,43] and interpenetrating hydrogels based on PNIPAm [35,38].

It is seen from Table 5 that *T*_2_ value of “confined” water in DNA1 hydrogel is relatively high in comparison with “confined” water in SN hydrogels and even in the case of other DN hydrogels with single-exponential relaxation decay, the occurrence of “confined” water molecules above the transition was not confirmed at all. This indicates that most of the permanently bound water molecules involved in collapsed deswollen regions is not significantly restricted in their mobility due to less intensive transition in comparison with SN hydrogels.

### 3.5. Release Properties

Time-dependent UV-Vis absorbance spectra of the methylene blue detected for all hydrogels are included in Appendix A, Appendix A. Figure 8 shows the release profile of MB molecules at 45 °C for hydrogels SN2, DNA2 and DNM2 (for data on all samples see Appendix A). It is seen that during the hydrogel collapse, MB molecules are released exponentially and time-dependences of release were fitted using exponential function according to Equation (4). The equilibrium fractions of released MB *f*_MB_ and the release delay time *τ* fitted from Equation (4) are for all hydrogels summarised in Table 6. Figure 8 and Table 6 show that the equilibrium release fractions are about 0.9 for SN hydrogels and 0.7–0.8 for DN hydrogels denoting high release efficiency of studied hydrogels. These values suggest that SN hydrogels with the most intensive temperature induced deswelling have the highest value of the equilibrium fraction of released MB, meanwhile DNM hydrogels release smaller fraction of MB molecules. Note that the low *f*_MB_ value for DNM2 hydrogel (Figure 8b) is also partly due to incomplete collapse of DNM2 network at 45 °C as can be seen in Figure 4 (only about 80% of bound water is released at 45 °C).

SN hydrogels release MB molecules in tens of minutes, but the release times registered for the both DNA and DNM hydrogels show values of one order of magnitude lower. Such a short release time could be a consequence of the hydrophilic character of DN hydrogels with the following mechanism. When swollen hydrogel sample (at room temperature) is suddenly submerged into the 45 °C water, the more hydrophobic SN hydrogel (in contrast to more hydrophilic DN hydrogels) can form a skin-like surface due to rapid collapse of the surface layer. This prevents remaining water from fast release. The formation of a hardly penetrable skin-like surface of SN hydrogels is also supported by the presence of a low amount of water in SN hydrogel above the phase transition temperature, as represented by the *β* parameter in Table 2 (obtained at quasi-equilibrium conditions).

## 4. Conclusions

Two series of DN hydrogels were prepared with various crosslinking densities of the first PDEAAm networks and constant (much lower) crosslinking density of the second PAAm or PDMAAm network, respectively. Swelling measurements, DSC and NMR and UV-Vis spectroscopies were combined to provide information about temperature-induced volume changes in hydrogels on macroscopic and molecular scales.

Presence of the second hydrophilic network in DN hydrogels has a strong impact on their thermal sensitivity. DN hydrogels show less intensive changes in deswelling, smaller enthalpy and entropy changes, and broader temperature interval of the transition compared to SN hydrogels. Above the phase transition temperature, the DN hydrogels contain significantly more permanently bound water in comparison with SN hydrogels due to the interaction of water with the hydrophilic component. Unlike swelling and DSC experiments, rather intensive transition was revealed from temperature-dependent NMR spectra as this detection is limited by reduced mobility of PDEAAm units in DN hydrogels. Release study showed that model MB molecules are released from SN and DN hydrogels within different time scale. In summary, temperature response of studied hydrogels was found to be tuneable by preparation parameters and this knowledge can be useful in the design of new responsive materials.

New thermodynamic model of deswelling behaviour based on the approach of the van’t Hoff analysis was developed. The model allows determining thermodynamic parameters connected with temperature-induced volume transition, such as the standard change of enthalpy and entropy and critical temperatures and characterize the structurally different states of water.

## Figures and Tables

**Figure 1 polymers-12-02502-f001:**
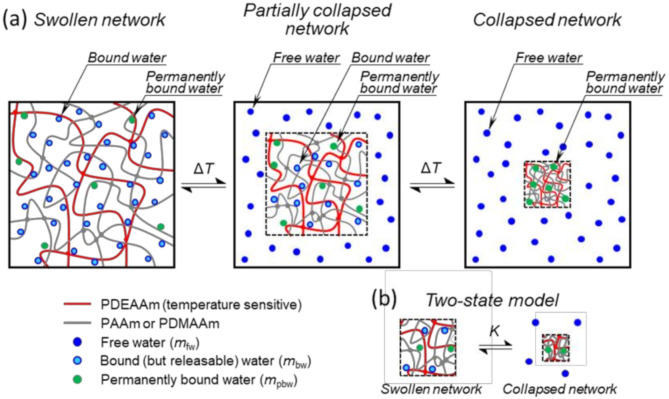
(**a**) Schematic model of temperature-dependent deswelling experiment of SN and double network (DN) networks (the figure shows an actual model of DN). The entire system is shown, and dashed-line squares denote the boundary of the polymer network during the collapse. (**b**) Two-state swollen-collapsed network model with permanently bound water.

**Figure 2 polymers-12-02502-f002:**
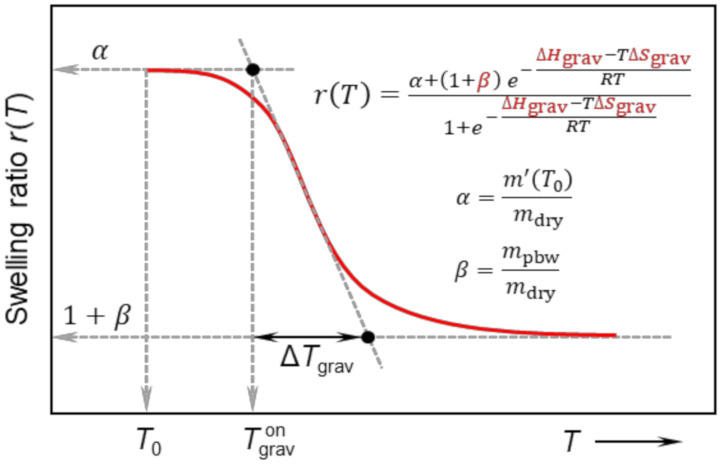
Theoretical dependence of swelling ratio *r*(*T*) as defined by Equation (16) (also present in the graph). The essential features of the curve are shown, including fitted parameters highlighted by red color.

**Figure 3 polymers-12-02502-f003:**
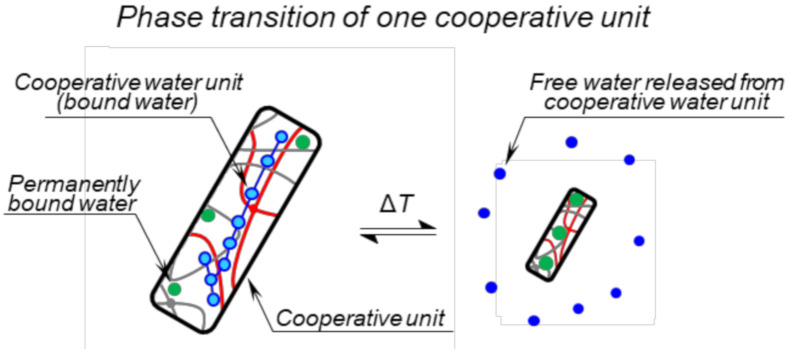
Schematic model of phase transition of one cooperative unit. Only one cooperative unit of the network sample is shown. Blue lines highlight the enhanced interaction between molecules of the cooperative water unit (containing *N*_wcu_ molecules).

**Figure 4 polymers-12-02502-f004:**
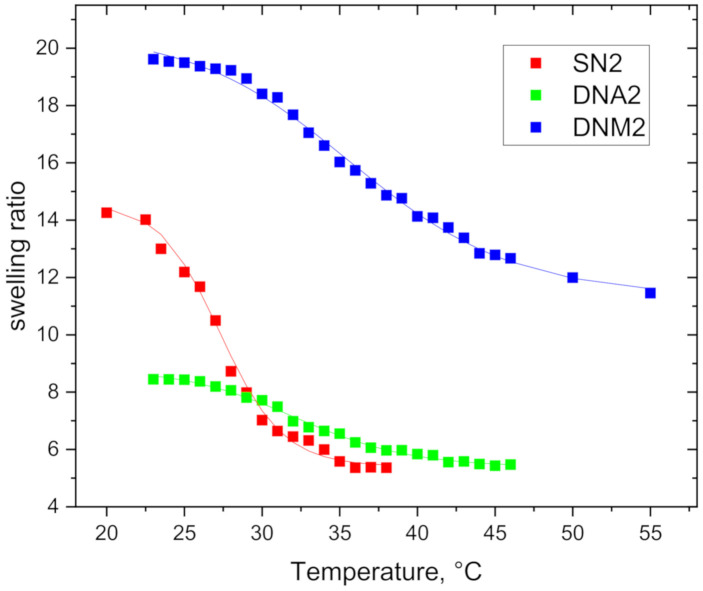
Temperature dependence of swelling ratio for hydrogels SN2, DNA2, and DNM2. Experimental points are fitted according to Equation (16).

**Figure 5 polymers-12-02502-f005:**
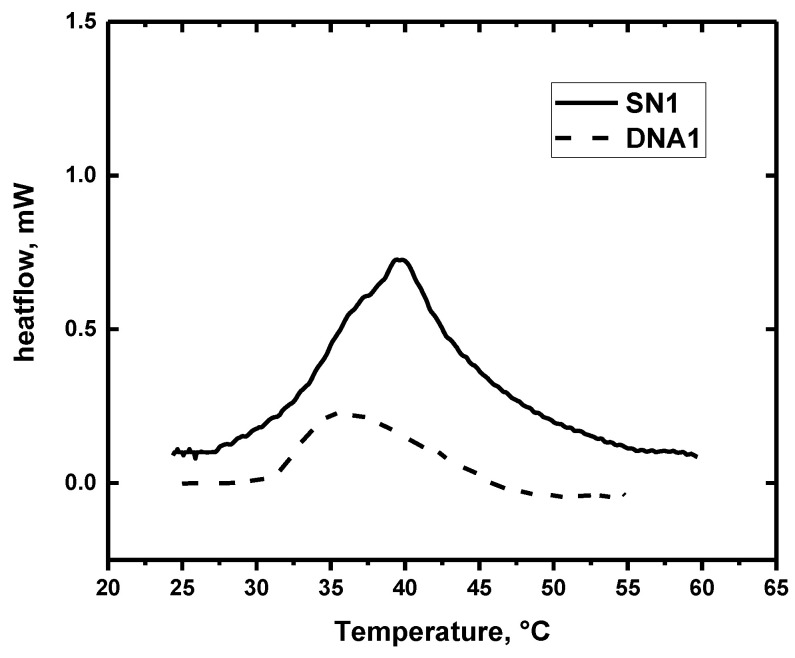
Differential scanning calorimetry (DSC) curves for SN1 and DNA1 hydrogels obtained during heating.

**Figure 6 polymers-12-02502-f006:**
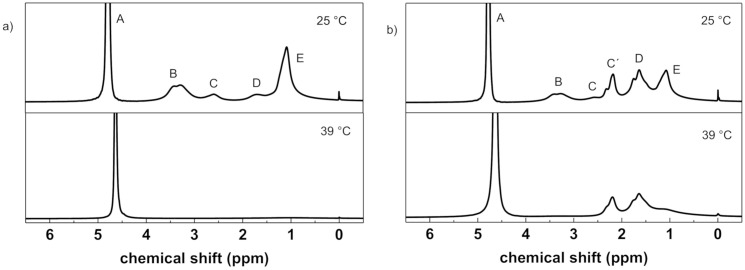
500.1 MHz ^1^H spectra of hydrogels (**a**) SN1 and (**b**) DNA1 in D_2_O recorded at 25 °C and 39 °C under the same instrumental conditions. Peak assignments are explained in the text.

**Figure 7 polymers-12-02502-f007:**
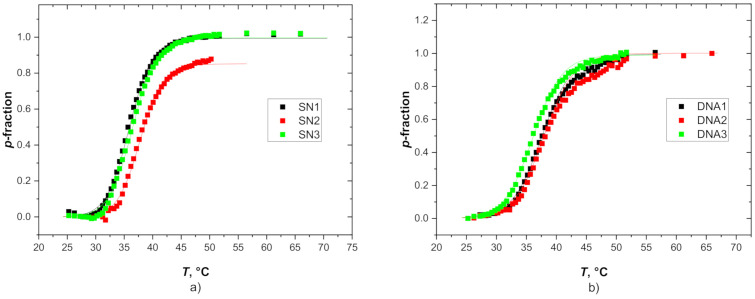
Temperature dependences of *p*-fraction as determined from PDEAAm CH_3_ signal in (**a**) SN and (**b**) DNA hydrogels. Experimental points are fitted according to Equation (2).

**Figure 8 polymers-12-02502-f008:**
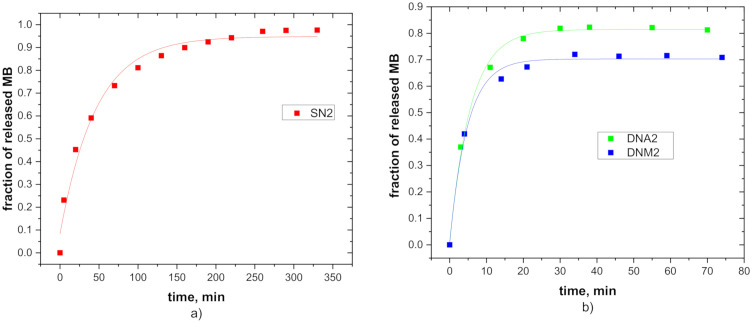
Time dependence of methylene blue (MB) concentration for release process for (**a**) SN2 hydrogel and (**b**) DNA2 and DNM2 hydrogels at 45 °C. Experimental points are fitted according to Equation (4).

**Table 1 polymers-12-02502-t001:** Chemical composition of reaction mixtures used in the preparation of single network (SN) hydrogels: mass concentrations of DEAAm (*c*_DEAAm_), MBAAm (*c*_MBAAm_), ammonium persulfate (APS) (*c*_APS_), and TEMED (*c*_TEMED_), and molar ratio of DEAAm and MBAAm in polymerization solution.

Sample	*c*_DEAAm_ (g.L^−1^)	*c*_MBAAm_ (g.L^−1^)	*c*_APS_ (g.L^−1^)	*c*_TEMED_ (g.L^−1^)	DEAAm/MBAAm
SN1	127.2	2.0	1	15	77
SN2 SN3	127.2 127.2	1.2 0.4	1 1	15 15	128 385

**Table 2 polymers-12-02502-t002:** Transition parameters of hydrogels determined during heating by gravimetric deswelling experiments: enthalpy Δ*H*_grav_, entropy Δ*S*_grav_, initial swelling ratio of sample *α = m’*(*T*_0_)/*m*_dry_, ratio of permanently bound water to mass of dry network *β* = *m*_pbw_/*m*_dry_, ratio of mass of permanently bound water to total mass of water *γ* = *m*_pbw_/*m*_0_, onset transition temperature $T_{\mathrm{grav}}^{\mathrm{on}}$ and width of transition $\Delta T_{\mathrm{grav}}$.

Sample	Δ*H*_grav_ (kJ.mol^−1^)	Δ*S*_grav_ (J.mol^−1^. K^−1^)	α	β	γ	$T_{grav}^{on}\;({}^{\circ}C)$	$\Delta T_{grav}\;({}^{\circ}C)$
SN1	246	832	11.1	2.93	0.150	21	10
SN2 SN3 DNA1 DNA2 DNA3 DNM1 DNM2 DNM3	284 289 211 199 152 141 115 90	954 971 694 652 505 460 374 295	14.6 21.5 6.98 8.81 14.5 17.6 20.5 33.7	4.41 4.28 2.70 4.37 9.12 7.93 10.3 18.7	0.162 0.098 0.453 0.539 0.672 0.450 0.429 0.572	23 24 25 25 24 28 26 24	8 8 14 15 19 16 21 28

**Table 3 polymers-12-02502-t003:** Transition characteristics of hydrogels determined during heating by DSC: specific enthalpy of demixing per unit mass of network sample Δ*H*_DSC-ms_, specific enthalpy of demixing per unit mass of polymer network Δ*H*_DSC-pn_, onset temperature of demixing $T_{\mathrm{DSC}}^{\mathrm{on}}$, peak temperature $T_{\mathrm{DSC}}^{\mathrm{peak}}$, and the number of water molecules in a cooperative water unit *N*_wcu_.

Sample	Δ*H*_DSC-ms_ (J.g^−1^)	Δ*H*_DSC-pn_ (J.g^−1^)	$T_{DSC}^{on}\;({}^{\circ}C)$	$T_{DSC}^{peak}\;({}^{\circ}C)$	*N* _wcu_
SN1 SN2 SN3 DNA1 DNA2 DNA3 DNM1 DNM2 DNM3	2.10 1.86 0.85 0.53 0.29 0.20 0.24 0.27 0.15	19 23 23 3.2 2.5 2.6 0.2 0.3 0.03	30 31 31 32 34 32 32 33 32	39 40 35 36 39 37 42 37 45	5300 5350 16,600 10,400 14,900 12,900 16,100 10,500 13,900

**Table 4 polymers-12-02502-t004:** Characteristic parameters as obtained from NMR spectra analyses: enthalpy Δ*H*_NMR_, entropy Δ*S*_NMR_, onset transition temperature $T_{\mathrm{NMR}}^{\mathrm{on}}$, width of transition Δ*T*_NMR_ and molar ratios of AAm (or DMAAm) and DEAAm *N*_A_*/N*_DEAAm_ (*N*_M_*/N*_DEAAm_).

Sample	Δ*H*_NMR_ (kJ.mol^−1^)	Δ*S*_NMR_ (J.mol^−1^.K^−1^)	$T_{NMR}^{on}\;({}^{\circ}C)$	Δ*T*_NMR_ (°C)	*N* _A_ */N* _DEAAm_	*N* _M_ */N* _DEAAm_
SN1 SN2 SN3 DNA1 DNA2 DNA3 DNM1 DNM2 DNM3	398 422 387 324 320 333 289 276 245	1289 1374 1251 1043 1028 1078 926 884 786	32 32 33 33 33 32 34 34 32	8 7 8 10 10 10 11 12 13	--- --- --- 2.4 2.7 4.9 --- --- ---	--- --- --- --- --- --- 2.0 2.2 3.6

**Table 5 polymers-12-02502-t005:** Spin-spin relaxation times *T*_2_ of hydrogen-deuterium oxide (HDO) protons for SN and DN hydrogels measured at 17 °C and 45 °C.

Sample	*T*_2_ (s)	
	17 °C	45 °C
SN1 SN2 SN3 DNA1 DNA2 DNA3 DNM1 DNM2 DNM3	2.8 4.4 4.1 4.3 3.9 3.0 3.8 3.1 3.7	2.2 * 0.1 * 4.8 * 0.03 * 5.1 * 0.04 * 3.9 * 0.9 * 3.3 2.7 4.1 3.8 4.3

* *T*_2_ values obtained from bi-exponential relaxation decay.

**Table 6 polymers-12-02502-t006:** Parameters related to MB release fitted from Equation (4): the equilibrium fractions of released MB *f*_MB_ and the release delay time *τ*.

Sample	*f* _MB_	*τ*
		(min)
SN1 SN2 SN3 DNA1 DNA2 DNA3 DNM1 DNM2 DNM3	0.93 0.96 0.91 0.85 0.81 0.89 0.69 0.71 0.76	64 59 55 9.5 7.2 9.1 8.3 8.4 12.3

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
