# Peer review of "Poly(N,N′-Diethylacrylamide)-Based Thermoresponsive Hydrogels with Double Network Structure"

_polymers, 2020, doi:10.3390/polym12112502_

Round 1
Reviewer 1 Report
In this work, Hanyková et al. studied the temperature response of double network (DN) hydrogels made from thermo-responsive poly(N,N’-diethylacrylamide) and hydrophilic polyacrylamide or poly(N,N’-dimethylacrylamide) by swelling measurements, DSC, NMR and UV-Vis studies. Their results showed the second network in DN gels has a significant impact on their thermo properties compared with single network gel. It is, to certain degree, similar to their previous paper (Eur. Polym. J. 2019, 116, 415– 424), but in this work, the authors used less toxic poly(N,N’-diethylacrylamide) instead of PNIPAM and added PDMAAm as a second components. In general, this topic is interesting but not novel, the authors tried to use different techniques to fully understand the materials and the discussion is good. The authors failed to provide most of the raw data (NMR, UV-Vis, DSC and swelling study, there are no supplementary materials).
Based on the current manuscript, detailed comments are listed below:
- The concentrations of AAm (142.2 g/L) and DMAAm (97.16 g/L) are different. How and why to choose the specific concentrations for the samples?
- How is the repeatability of the experiments (swelling behavior, DSC, NMR and UV-Vis results)?
- Page 13, Line 409. The authors assumed most of the PDEAAm units is detected by NMR. One suggestion to confirm such assumption would be use of an internal/external standard to quantitatively determine how much of PDEAAm units were detected during the measurements.
- Figure 7b, some values of the p-fraction are larger than 1. This should not be possible from definition. How is the NMR baseline in the spectra?
- Page 15, Line 449. When discussing the short release time of DN gels, the authors claimed one of the reasons could be the more prevailing hydrogen bonding between water molecules and polymer in DN gel, compared to SN gel. However, the data from entropy change (ΔSNMR, table 4) indicates the opposite way: the entropy changes for SN samples are more positive than the DN samples, for example, 1289 J/(mol K) for SN1 gel and 926 J/(mol K) for DNM1 gel. More positive entropy change means more water loss for that sample, which indicates that SN1 gels can bind more water, or stronger hydrogen bonding, than DN gels.
- The abstract is longer than the requirement.
- I would suggest to use name “SN1” “SN2”, “SN3” instead of “SN1”, “SN3”, “SN5”, except there is a special reason.
Author Response
The response to Reviewer 1 is attached.

Reviewer 2 Report
The paper” Poly(N,N`-diethylacrylamide)-based thermoresponsive hydrogels with double network structure by Lenka Hanyková, Ivan Krakovský, Eliška Šestáková, Julie Šťastná and Jan Labuta (Polymers 967934) deals with the study of thermoresponsive hydrogels of poly(N,N`-diethylacrylamide) (PDEAAm) (SN) and its double network (DN) with hydrophilic polyacrylamide (PAAm) or poly(N,N`-dimethylacrylamide) (PDMAAm). The results obtained gravimetrically, by DSC and 1H NMR are discussed. The model compound release (methylene blue) from the networks is reported.
The concept of the work is not new. Similar materials i.e. IPNs of PDEAAm and PAAm [36] prepared according to the same procedure as in the current paper and DN of PNIPAM/PAAm [37] were studied by the Authors previously. The thermodynamic model of deswelling is newly developed.
The paper is interesting but the authors omitted many important issues in the discussion. The authors should elaborate on two issues essential for a correct comparison of the obtained networks,
- the relationship between the obtained parameters and the structure/composition of the network The composition of the network is not defined sufficiently because of process of preparation and as the authors claimed the behavior of SN of PDEAAm below the polymer VPTT.
- An explanation of the difference in the observed parameters caused by the chemical structure/amount of the second polymer (PAAm, PDMMm) in DN networks.
A significant weakness of work is the lack of a clear reference to the results described in previous papers [36] and [37]. Authors should indicate similarities or differences in the results for similar materials tested.
Comments on the results presented in the study and the examples of misleading descriptions are given below
Examples:
Abstract, strange wording:
DN hydrogels show less intensive changes in deswelling, smaller enthalpy and entropy variations and broader temperature interval of the transition. LESS THAN WHAT?
SN abbreviation undefined in the abstract
In: 3.1. Swelling behaviour
The lack of the explanation of the dependences given in Fig. 4. Why more hydrophobic DNM (two CH3 groups in PDAAm) swells 2.5 times more than DNA.
r(T) vs temperature dependences for two other sets of results should be shown in Supplementary. (Supplementary will be helpful for the readers).
The paragraph between 301 and 307 should be rewritten, it is not clear.
“It is obvious from Table 2 that DN hydrogels show smaller enthalpy and entropy variations of the transition,” – what “variation” is it about?
In: 3.2. Differential scanning calorimetry
The paragraph between 343 and 349 should be rewritten indicating clearly the type of hydrogel discussed
In: 3.3. NMR
In the paragraph 396 to 413 the Authors try to discuss the structure and the network composition. The conclusions they draw are inconsistent or wrongly formulated. Especially “These immobile units do not contribute to high resolution 1H NMR spectra even at temperatures below transition which decreases NMR detected molar ratio of AAm or DMAAm and DEAAm in DN hydrogels NA/NDEAAm or NM/NDEAAm, respectively” is incomprehensible. if the denominator decreases, the value of the fraction increases as it seems to be.
In: 3.5. Release properties
“Fig. 8 shows the release profile of MB solution” – MB not MB solution
Fig. 8 inappropriate description
The influence of the second co-network on the release profile (fig. 8b) is confusing when the relation in fig 4 is taken into account (higher swelling of DMN comparing to DNA). It should be explained.
Some language errors can be found.
Author Response
The response to Reviewer 2 is attached.

Round 2
Reviewer 1 Report
Most responses are fine, expect:
(1) the authors still failed to provide raw data (Figures S7-S10 are not raw data);
(2) the reproducibility cannot be claimed without showing any data (Lines 116-118, all experiments were performed on both specimens to probe repeatability and reliability of obtained characteristic parameters).
Reviewer 2 Report
The paper was improved sufficiently. Typing error is present in the conclusions.
Author Response
Typing errors were corrected.
Round 3
Reviewer 1 Report
As the authors added corresponding raw data into SI, the manuscript is now recommended to be published in Polymers.